# Proteomic Analysis of Arsenic Resistance during Cyanide Assimilation by *Pseudomonas pseudoalcaligenes* CECT 5344

**DOI:** 10.3390/ijms24087232

**Published:** 2023-04-13

**Authors:** Karolina A. Biełło, Purificación Cabello, Gema Rodríguez-Caballero, Lara P. Sáez, Víctor M. Luque-Almagro, María Dolores Roldán, Alfonso Olaya-Abril, Conrado Moreno-Vivián

**Affiliations:** 1Departamento de Bioquímica y Biología Molecular, Edificio Severo Ochoa, Campus de Rabanales, Universidad de Córdoba, 14071 Córdoba, Spain; karolinabiello@gmail.com (K.A.B.); b42lualv@uco.es (V.M.L.-A.); bb2rorum@uco.es (M.D.R.); bb1movic@uco.es (C.M.-V.); 2Departamento de Botánica, Ecología y Fisiología Vegetal, Edificio Celestino Mutis, Campus de Rabanales, Universidad de Córdoba, 14071 Córdoba, Spain; bv1cahap@uco.es

**Keywords:** arsenic resistance, biofilm, cyanide assimilation, quantitative proteomics, *Pseudomonas pseudoalcaligenes*

## Abstract

Wastewater from mining and other industries usually contains arsenic and cyanide, two highly toxic pollutants, thereby creating the need to develop bioremediation strategies. Here, molecular mechanisms triggered by the simultaneous presence of cyanide and arsenite were analyzed by quantitative proteomics, complemented with qRT-PCR analysis and determination of analytes in the cyanide-assimilating bacterium *Pseudomonas pseudoalcaligenes* CECT 5344. Several proteins encoded by two *ars* gene clusters and other Ars-related proteins were up-regulated by arsenite, even during cyanide assimilation. Although some proteins encoded by the *cio* gene cluster responsible for cyanide-insensitive respiration decreased in the presence of arsenite, the nitrilase NitC required for cyanide assimilation was unaffected, thus allowing bacterial growth with cyanide and arsenic. Two complementary As-resistance mechanisms were developed in this bacterium, the extrusion of As(III) and its extracellular sequestration in biofilm, whose synthesis increased in the presence of arsenite, and the formation of organoarsenicals such as arseno-phosphoglycerate and methyl-As. Tetrahydrofolate metabolism was also stimulated by arsenite. In addition, the ArsH2 protein increased in the presence of arsenite or cyanide, suggesting its role in the protection from oxidative stress caused by both toxics. These results could be useful for the development of bioremediation strategies for industrial wastes co-contaminated with cyanide and arsenic.

## 1. Introduction

The biogeochemical cycle of arsenic (As), a metalloid widely distributed in the lithosphere, encompasses both natural and anthropic processes [1,2,3,4]. Due to human activities, arsenic can reach molar range concentrations in gold mining effluents [5], where other toxic pollutants such as cyanide and heavy metals are also present. Arsenic is considered a serious environmental pollutant that is highly toxic to humans [6,7,8]. In fact, arsenic-contaminated groundwater caused the largest poisoning in human history in Bangladesh [9].

Microbes may transform arsenic into different inorganic and organic forms [10]. Trivalent arsenite and pentavalent arsenate are the most abundant inorganic forms, with As(III) being 100 times more toxic than As(V) [4]. Arsenate toxicity is mainly derived from its ability to compete with phosphate oxyanions for both transport and energetic functions [11], whereas arsenite binds to sulfhydryl groups of proteins and biomolecules such as glutathione and lipoic acid [1]. Both forms enter adventitiously into the cells through different uptake systems [12]. Thus, arsenate is a phosphate analog that uses the phosphate permeases, whereas the internalization of arsenite may occur through aquaglyceroporins, such as GlpF in *E. coli* and AqpS in *Sinorhizobium meliloti* [13]. Organic arsenical compounds, such as monomethy larsenate (MMAV), dimethy larsenate (DMAV), and trimethylarsine (TMA), also occur in nature. These methyl arsenicals are usually less toxic than their inorganic counterparts, except for the highly toxic trivalent derivatives monomethy larsenite (MMAIII) and dimethylarsenite (DMAIII) [14].

Ancient and ongoing exposure to arsenic has allowed the evolution or acquisition of various arsenic resistance systems in microorganisms, but with different degrees of efficiency [15,16]. A common strategy for tolerating inorganic arsenic is its extrusion from the cytosol [12,17], although other mechanisms have been also described, such as intracellular chelation by metal-binding peptides [18], bioaccumulation [19], immobilization, and transformation to potentially less toxic organic forms [20]. Prokaryotes show the greatest metabolic diversity with respect to arsenic metabolism, which is the detoxification pathway encoded by the *ars* genes most widely distributed among bacteria and archaea [16,21,22,23,24]. The As-resistance systems (Ars) require at least three components: an arsenite-responsive regulatory protein (ArsR), an efflux pump (ArsB), and an arsenate reductase (ArsC), although other *ars* genes are also included in some microbial *ars* gene clusters [25]. ArsR is a repressor that, after binding arsenite, dissociates from the promoter region to allow transcription of structural *ars* genes [26]. ArsB is a transmembrane protein that mediates arsenite export to the extracellular media [27]. The ATPase ArsA may form a complex with ArsB to yield a high-efficiency ATP-driven efflux pump. Additionally, the permease Acr3 may be present or replace the ArsB efflux pump [28]. The arsenate reductase ArsC transforms As(V) into As(III), which is extruded by the efflux pump. Depending on the electron donor used, two unrelated types of arsenate reductases are found, glutaredoxin- and thioredoxin-dependent ArsC [16,23].

Microbial formation of organoarsenicals, mainly by methylation, has also an important role in the arsenic biogeochemical cycle [16,21,23]. Arsenic methylation is usually considered a detoxification mechanism, but when toxic methyl-As(III) derivatives such as MMAIII or DMAIII are formed, demethylation may be part of the detoxification process [29]. Bacterial organoarsenical detoxification proteins include the *S*-adenosylmethionine (SAM)-dependent methyltransferase ArsM, which may have both methylating and demethylating activity, the NADPH-dependent FMN reductase ArsH, a protein that reduces ferric and chromate ions and shows methyl arsenite oxidase activity to confer resistance to methyl-As(III) derivatives, but also protects from oxidative damage caused by exposure to As [25], the permease ArsP that confers tolerance to MMAIII and DMAIII, and the non-heme iron-dependent dioxygenase ArsI, which shows C-As lyase activity [30]. Arsenic resistance in *Pseudomonas aeruginosa* DK2 involves the *gapdh* and *arsJ* gene products. The glyceraldehyde-3-phosphate dehydrogenase GAPDH uses arsenate, an inorganic phosphate analog, to form 1-arseno-3-phosphoglycerate, and this unstable organoarsenical compound is extruded by ArsJ, a major facilitator superfamily (MFS) permease, before its dissociation into As(V) and 3-phosphoglycerate [31].

Arsenic and cyanide are usually co-contaminants of industrial wastewater from mining and other industries. Environments co-contaminated with different metals or chemicals are less understood, and more difficult to remediate than single-polluted systems [32]. Although As-resistance has been well characterized in different bacteria, the capability to tolerate and detoxify arsenic in the presence of cyanide has not been studied yet. In this work, the first holistic approach combining quantitative proteomics and qRT-PCR transcriptional analysis of relevant genes was carried out to characterize arsenic and cyanide detoxification in *Pseudomonas pseudoalcaligenes* CECT 5344, a bacterium isolated from the sludges of the Guadalquivir River (Córdoba, Spain) that can grow with cyanide, cyanate, 3-cyano alanine and other nitriles, and industrial residues containing cyanide and metals [33,34,35,36]. In this strain, the cyanide-insensitive respiration and the cyanide assimilation pathway are linked. Thus, the alternative oxidase CioAB encoded by the *cio* gene cluster is coupled to malate:quinone oxidoreductase that oxidizes malate to oxalacetate, which reacts with cyanide forming a 2-hydroxy nitrile. This nitrile is further hydrolyzed by the nitrilase NitC encoded by the *nit1C* gene cluster, generating ammonium used as a nitrogen source for growth [34]. The genome of this strain was completely sequenced [37,38], and omic studies on the cyanide assimilation were previously performed [39,40,41,42,43,44], making this bacterium a suitable model to develop system and synthetic biology strategies for bioremediation of residues co-contaminated with cyanide and arsenic [45]. Knowledge of the mechanisms involved in the resistance and detoxification of arsenic and cyanide is essential for the bioremediation of mining wastes and systems polluted with both contaminants.

## 2. Results and Discussion

### 2.1. Bioinformatic Analysis of Arsenic Detoxifying Potential in P. pseudoalcaligenes CECT 5344

The presence in the P. pseudoalcaligenes CECT 5344 genome of putative genes involved in arsenic tolerance was analyzed by protein-BLAST using available information from sequences deposited in UniProt. Genes coding for three arsenate reductases (ArsC), two arsenite efflux pumps (ArsB), two regulatory proteins (ArsR), and other putative proteins involved in arsenic detoxification were identified in two different *ars* gene clusters (Figure 1A), but other *ars* genes were also distributed throughout the genome. The *ars* gene cluster 1 of P. pseudoalcaligenes CECT 5344 showed the gene arrangement arsRICBCH (BN5_1989–BN5_1994), with two genes coding for different arsenate reductases, the thioredoxin-dependent ArsC1 (BN5_1991), and the glutaredoxin-dependent ArsC2 (BN5_1993). The *ars* gene region 2, comprising from BN5_2705 to BN5_2712, displayed the arsABRCH gene organization, in which the arsC gene (BN5_2708) codes for a thioredoxin-dependent arsenate reductase (ArsC3). Additionally, the *gapdh* (BN5_2711) and *arsJ* (BN5_2712) genes were included in this *ars* gene arrangement, which code for a glyceraldehyde-3-phosphate dehydrogenase and an organoarsenical MSF-type transporter, respectively (Figure 1A). In addition to these *ars* gene regions, the genome of P. pseudoalcaligenes CECT 5344 includes two other arsC genes (BN5_2838 and BN5_2786) for putative glutaredoxin-dependent arsenate reductases (ArsC4 and ArsC5, respectively), and genes coding for an ArsR-family transcriptional regulator (BN5_0252), an Acr3-type permease/multidrug resistance protein MdtC (BN5_2698), and a multi-transmembrane protein (BN5_1279) that contains an ArsP domain according to the Pfam protein family database. In addition, several predicted proteins of the strain CECT 5344 showed similarity (40–50%) with bacterial ArsM methyltransferases, mainly aligned in the central methylase domain. These ArsM-like proteins are annotated in the P. pseudoalcaligenes CECT 5344 genome as 3-demethylubiquinone-9 3-methyltransferase (BN5_1671), bifunctional demethylmenaquinone methyltransferase/2-methoxy-6-polyprenyl-1,4-benzoquinol methylase UbiE (BN5_0407), quinone oxidoreductase (BN5_0233), ubiquinone/menaquinone biosynthesis methylase (BN5_3586), biotin synthesis protein BioC (BN5_3914), and uncharacterized protein (BN5_4176). However, homologs to the aio, arx, or arr genes, which are involved in As(III) oxidation or dissimilatory As(V) reduction, were not found in the genome of P. pseudoalcaligenes CECT 5344.

### 2.2. Arsenic Resistance of P. pseudoalcaligenes CECT 5344

The minimal inhibitory concentration (MIC) and the minimal bactericidal concentration (MBC) were calculated by growing the strain CECT 5344 with 2 mM ammonium chloride or 2 mM sodium cyanide as the sole nitrogen source in the presence of different As(III) or As(V) concentrations. For As(III), the MIC and MBC values were 35 and 40 mM, respectively, regardless of the N-source used (ammonium or cyanide). These values are similar to those found in Bacillus licheniformis DAS1 [46] or in several bacterial isolates from tannery wastes [47]. Interestingly, the MIC and MBC values for cyanide, which were 25 and 50 mM in absence of arsenite, dropped to 20 and 25 mM, respectively, in the presence of 0.25 mM arsenite, thus indicating that As(III) decreases cyanide tolerance of P. pseudoalcaligenes CECT 5344. This fact should be considered for the bioremediation of wastes co-contaminated with both toxics. Regarding As(V), this bacterium showed MIC values of 1.25 M (with cyanide as N-source) or 1.75 M (with ammonium), and MBC values of 2 M (with cyanide) or 2.5 M (with ammonium), in accordance with the lower toxicity of arsenate compared to arsenite described by other authors [48,49,50].

Before performing the quantitative proteomics and qRT-PCR analyses, a physiological characterization was carried out to determine an arsenite concentration that could affect cell growth and, therefore, metabolism without completely inhibiting bacterial growth. P. pseudoalcaligenes CECT 5344 was grown with different concentrations of arsenite (up to 1 mM), using either ammonium or cyanide as the sole nitrogen source (Figure 1B). It was observed that, with both nitrogen sources, a concentration of 0.25 mM of arsenite, almost 10 times higher than the average concentration of arsenic in the lithosphere [51], conditioned cell growth sufficiently without being highly detrimental to bacterial growth (Figure 1B). Likewise, P. pseudoalcaligenes CECT 5344 was highly resistant to As(V), but it was observed that 200 mM arsenate negatively affected bacterial growth (Appendix A). Therefore, 0.25 mM arsenite and 200 mM arsenate were used for the subsequent analyses.

### 2.3. Global Changes in the Proteome of P. pseudoalcaligenes CECT 5344 in Response to Arsenite

To study the general response against As(III) in this bacterium, quantitative proteomic analysis by LC-MS/MS was carried out using cells grown under four conditions: 2 mM ammonium chloride as the sole nitrogen source, without or with 0.25 mM arsenite addition (conditions N or NAs, respectively), and 2 mM sodium cyanide as sole nitrogen source, without or with 0.25 mM arsenite (conditions CN or CNAs, respectively). The samples were collected before exhausting the nitrogen source to prevent a nitrogen starvation response. The three biological replicas carried out for each condition were grouped according to both the nitrogen source and the presence or absence of arsenite (Appendix A). A total of 1659 unique proteins were identified from more than one peptide: 1177 in N, 1312 in NAs, 1591 in CN, and 1581 in CNAs (Appendix A), although most of them (1105 proteins; 66.6% of identified proteins) were shared among all conditions (Appendix A).

In the differential expression analyses, the comparisons of CNAs vs. CN and NAs vs. N revealed the protein expression changes that are due to the presence of arsenite with both nitrogen sources (Appendix A). The comparison CNAs vs. CN revealed that 24 proteins were up-regulated in the CNAs cultures, of which 17 were ‘exclusive’ (that is, found in at least two replicas of the condition CNAs and not detected in CN), and 7 were over-represented (fold change FC ≥ 2) in the presence of As(III). Conversely, 41 proteins were up-regulated in the CN condition, with 18 proteins exclusive and 23 proteins over-represented in the absence of arsenite. The comparison of NAs vs. N showed that 87 proteins were up-regulated by arsenite in the NAs cultures, of which 86 were exclusive (not found in N), and only one protein was over-represented in NAs with respect to N, whereas 18 proteins were exclusive of the N cultures (not detected in NAs).

The differential proteomic analyses CN vs. N and CNAs vs. NAs reflected how the bacterium responds to cyanide in the absence or presence of arsenite, respectively. The analysis CN vs. N showed 353 proteins exclusive plus 188 proteins over-represented in CN, and 18 proteins exclusive plus 164 proteins over-represented in N. Comparing CNAs vs. NAs, 241 proteins were found exclusively in CNAs, and 168 proteins were over-represented, whereas 23 proteins were exclusive and 137 proteins were over-represented in NAs (SDS2).

Results obtained in the differential proteomic analysis CN vs. N presented a high degree of overlap with omic results previously described in this cyanide-degrading strain [39,41,43,44], thus supporting the robustness and reliability of the analysis. The four nitrilases (NitC, Nit4, Nit1, and Nit2) that are present in P. pseudoalcaligenes CECT 5344 [37] were identified in the proteomic analysis (Table 1). As previously described, cyanide up-regulated the nitrilases NitC and Nit4, which are required for cyanide and 3-cyanoalanine assimilation, respectively [34,44]. NitC is encoded by the nitC (BN5_1632) gene of the nit1C cluster, whereas nit4 (BN5_1912) belongs to the cio gene cluster involved in the cyanide-insensitive respiration [37]. NitC was not significantly affected by arsenite (FC 0.91 in CNAs/CN), but the other two proteins encoded by the nit1C gene cluster, NitD, and NitG, were decreased in the CNAs condition. Similarly, Nit4 and most proteins encoded by the cio gene cluster were up-regulated by cyanide but decreased significantly in the presence of arsenite, showing FC values lower than 0.5 in CNAs/CN (Table 1). This may explain the drop in the MIC and MBC values for cyanide observed when the strain CECT 5344 was grown in the presence of arsenite 0.25 mM. Induction by arsenite, regardless of the nitrogen source used, was observed for the phosphate permease PstA, the protein PilJ involved in type IV pili formation, mobility and biofilm synthesis [52], the multidrug resistance/Acr3-type permease MdtC, the glycerol-3-phosphate dehydrogenase GlpD1, the protein translocase component SecY, and a LuxR family regulator, whereas the DNA repair protein RecN, the diguanylate cyclase YddV, and the membrane transport ATPase component ZntA, among other proteins, were exclusive of CNAs (Table 1). Interestingly, when comparing proteins up-regulated by cyanide in the absence of arsenite (CN vs. N) with those induced by arsenite in the absence of cyanide (NAs vs. N), it was found that 77 exclusive proteins were shared between the CN and NAs proteomes (SDS2). Some of these proteins are related to oxidative stress, such as the glutathione S-transferase and the hydrogen peroxide-inducible activator encoded by the BN5_4225 gene; mobility, cell communication, and biofilm formation, such as FlgN and PilJ proteins; cyanide metabolism such as the aliphatic nitrilase Nit1; and arsenic metabolism such as the ArsA and ArsH2 proteins, among others (SDS2). Therefore, cyanide and arsenic detoxification processes show a certain degree of interrelation, being affected by each other, probably because both toxics trigger general stress response mechanisms.

Comparative gene ontology (GO enrichment) was carried out to analyze the arsenite effect on global metabolism. In general, the response to arsenite was influenced by the nitrogen source (Appendix A). Thus, in ammonium-grown cells, proteins up-regulated by As gave an enrichment in the GO categories SOS response proteins, nucleoside metabolic process related proteins, and aromatic amino acid family biosynthetic process/chorismate biosynthetic process proteins. On the other hand, the response to arsenite in the presence of cyanide triggered non-specific responses related to nitrate assimilation and redox processes through the down-regulation of the assimilatory nitrate and nitrite reductases, and the cytochrome c oxidase cbb3-type CcoG, among other proteins. Arsenite may provoke changes in the metabolism of nitrogen compounds, as suggested by the induction of the glutamate synthase, the N-carbamoyl-L-amino acid amidohydrolase, and the molybdopterin molybdochelatase MoeA and siroheme synthase CysG proteins involved in the biosynthesis of the molybdopterin and siroheme cofactors of the nitrate and nitrite reductases, respectively (Table 1). Regarding other metabolic processes, the induction of enzymes involved in tetrahydrofolate (THF) metabolism was also observed in the arsenic cultures NAs and CNAs (Figure 2).

Concerning As metabolism, the differential proteomic analyses NAs vs. N and CNAs vs. CN revealed the metabolic strategy followed by P. pseudoalcaligenes to resist and detoxify arsenic. According to proteomic data, the *ars* gene clusters 1 and 2 responded significantly to the presence of arsenite. Thus, several proteins encoded by these gene clusters, such as the arsenate reductases ArsC1 and ArsC2, the pump-driving ATPase ArsA, the regulatory protein ArsR2, and the glyceraldehyde-3-phosphate dehydrogenase GAPDH, among other proteins, were up-regulated by arsenite (Table 1). In contrast, the putative arsenate reductases encoded by the non-clustered arsC genes BN5_2786 (ArsC5) and BN5_2838 (ArsC4) were not detected in the proteomic analysis.

Arsenite also induced the phosphate permease PstA, an arsenate transporter (Table 1). Once inside the cells, As(V) could be reduced to As(III) by the arsenate reductases ArsC1-C3, with arsenite directly expelled to the extracellular media by the ATP-driven efflux pump ArsBA or by other permeases such as ArsP or the multidrug resistance Acr3-type permease, which are also induced by As (Table 1). Although ArsB proteins were not identified by proteomics, probably because of technical limitations due to its transmembrane location [53], the ArsA component of the ArsBA efflux pump, and other proteins encoded by the *ars* clusters 1 and 2, were up-regulated by arsenite according to proteomic data (Table 1, Figure 1A).

Intracellular As bioaccumulation is not common in prokaryotes [54]. Accordingly, the induction of proteins related to this mechanism was not observed in the CECT 5344 strain. However, the synthesis of organoarsenic compounds, which later would be expelled from the cells, also contributes to arsenic detoxification in bacteria [21,23,55]. The proteomic analysis revealed that some ArsM-like proteins were constitutively expressed or induced by cyanide, suggesting that As(III) methylation may occur in P. pseudoalcaligenes CECT 5344, although these putative methyltransferases were not directly regulated by arsenite. Proteins related to THF metabolism (Figure 2) and also proteins involved in the synthesis of chorismate, a precursor of p-aminobenzoate, were induced in the presence of As(III), suggesting a relationship between THF metabolism and arsenic resistance. It has been shown that the administration of folic acid reduces arsenic toxicity in humans [56], and this effect could be due to the requirement of THF for arsenic methylation, either directly or allowing the synthesis and recycling of SAM, which is required for ArsM-type methyltransferases [57,58]. In addition, the enzyme GAPDH catalyzes the formation of arseno-phosphoglycerate, which is exported by the permease ArsJ [30,31]. In P. pseudoalcaligenes CECT 5344, both *gapdh* (BN5_2711) and *arsJ* (BN5_2712) genes are present in the arsenite-induced *ars* cluster 2 (Figure 1A). In addition, the glycerol-3-phosphate dehydrogenase GlpD1 was induced in both NAs and CNAs conditions (Table 1). On the other hand, nitriles and CN derivatives may react with arsenic forming Lewis adducts [59]. Dissociation of these possible adducts could be mediated by the putative C-As lyase ArsI encoded by the BN5_1990 gene of the *ars* cluster 1, which was found to be induced by arsenite (Table 1). Interestingly, the ArsH2 protein encoded by the BN5_2709 gene of the *ars* cluster 2 was up-regulated by arsenite and cyanide in the absence of arsenite, being exclusive to the CN condition in the CN vs. N comparison (Table 1). It has been recently described that ArsH2 contributes to relieving As toxicity in Pseudomonas putida, protecting from oxidative damage caused by exposure to arsenic and diamide [25]. Therefore, ArsH2 could be also involved in the defense against the oxidative stress caused by both arsenic and cyanide in P. pseudoalcaligenes CECT 5344.

### 2.4. Accumulation of Arsenic in Biofilm

It has been recently reported that As may be retained by the bacterial biofilm [17,55,60]. In addition, some proteins such as PilJ (BN5_0312), RfbA (BN5_4136), and RfbC (BN5_4137), which are involved in biofilm formation [52,61,62] were found to be up-regulated by arsenite in the proteomic analysis (Table 1, SDS2). Therefore, the capacity to synthesize biofilm and the possible accumulation of arsenic inside the cells or in the biofilm was determined. Biofilm formation was significantly increased in cells grown with arsenite, being the CNAs condition where the highest amount of biofilm was produced (Figure 3A). On the other hand, it was also observed that arsenic was retained mainly in biofilm, and intracellular accumulation of arsenic was residual. The accumulation of arsenic in biofilm was also significantly higher in the CNAs condition than in NAs (Figure 3B). These results indicate that As is extruded from the cytoplasm to the extracellular media, being retained and accumulated in the cell biofilm.

### 2.5. Transcriptional qRT-PCR Analysis

A qRT-PCR transcriptional analysis of some genes coding for relevant proteins identified by proteomics was also performed. First, the arsenate reductase genes of the strain CECT 5344 were analyzed in cells grown in the presence of 0.25 mM arsenite or 200 mM arsenate, with ammonium or cyanide (2 mM) as the sole nitrogen source. The expression of the arsenate reductases *arsC1* and *arsC2* genes (BN5_1991 and BN5_1993, respectively), located in the *ars* cluster 1, as well as *arsC3* (BN5_2708), located in the *ars* cluster 2, were induced by As(III) and As(V) regardless of the nitrogen source, cyanide, or ammonium (Figure 4). However, the expression of the putative genes *arsC5* (BN5_2786) and *arsC4* (BN5_2838) was residual and resulted in being unaffected by arsenite or arsenate (not shown). Therefore, proteomic and qRT-PCR data are coincident and suggest that the arsC1-C3 genes are involved in As-resistance, but the *arsC4* and *arsC5* genes could be the result of gene duplications or horizontal gene transfer events, being either non-functional or controlled by different environmental conditions, such as what occurs in *Pseudomonas putida* [63] and *Rhodopseudomonas palustris* [64].

In general, the proteomic results were confirmed when the expression of selected genes was analyzed by qRT-PCR (Appendix A). Thus, qRT-PCR analysis showed that the expression of the arsP gene (BN5_1279) encoding a putative permease and the gcvH3 (BN5_3016) and glyA5 (BN5_3018) genes, which are related to THF biosynthesis (Figure 2), were induced by arsenite. Increases of both protein and mRNA in the presence of arsenite were also observed for the phosphate permease PstA. Arsenite induction of the arsB1 (BN5_1992) and arsR1 (BN5_1989) genes (*ars* cluster 1) was observed only at the transcriptional level (Appendix A). This may be due to the difficulty to detect proteomics membrane proteins or regulators that may be present in the cells at very low concentrations [53]. On the other hand, the negative effect of arsenite on the cio gene cluster (cioA3 and maeB3 genes) was confirmed by qRT-PCR. For the maeB3 gene, this inhibition was observed both by transcriptional and proteomic analyses, but for the cioA3 gene, it was observed only at the protein level. Proteomic and qRT-PCR data revealed that the nitrilase NitC, which is essential for cyanide assimilation [34], was not affected by arsenite (Appendix A), thus allowing bacterial growth in the presence of both toxins, cyanide, and arsenite. However, the proteins NitD and NitG, which are also encoded by the nit1C gene cluster were found to decrease in the presence of arsenite (Table 1). Significant differences were not observed at the transcript level for nitD and nitG in the CN and CNAs conditions (Appendix A), suggesting that As(III) affects NitD and NitG only post-translationally.

### 2.6. Overview of the As Resistance Mechanisms in P. pseudoalcaligenes CECT 5344

Considering all the results of this work, the arsenic metabolism in P. pseudoalcaligenes CECT 5344 could be summarized as shown in Figure 5. Arsenite induces the expression of the *ars* gene clusters 1 and 2, probably through the inactivation of ArsR. Arsenate enters the cells through the PstA phosphate permease, and then is reduced to As(III) by the arsenate reductases ArsC1-C3. The main mechanism of arsenite tolerance is its extrusion through the ArsBA-type efflux pump. The formation of organoarsenical compounds also contributes to As detoxification. Methylated derivatives could be formed by methyltransferases that show similarity to bacterial ArsM proteins, and proteomic data suggest that THF metabolism may play a role in the methylation processes. Moreover, an arsenic-phosphoglycerate derivative may be formed by the arsenite-inducible glyceraldehyde-3-phosphate dehydrogenase GAPDH. Organoarsenicals could be extruded by ArsJ, Acr3, and ArsP permeases. Once outside, arsenic is retained and accumulated on the biofilm. Moreover, when cyanide is used as the nitrogen source, arsenite and cyanide could form Lewis adducts, which could be processed by ArsI. Additionally, the protein ArsH2, which is induced by arsenite or cyanide, could increase the tolerance to these compounds by protecting them from oxidative stress caused by these toxins. Results derived from this work may contribute to the development of strategies for the bioremediation of wastes containing cyanide and arsenic as co-pollutants by using the P. pseudoalcaligenes CECT 5344 strain and taking into account that cyanide and arsenic detoxification processes affect each other.

## 3. Materials and Methods

### 3.1. Culture Media and Growth Conditions

*P. pseudoalcaligenes* CECT 5344 was cultured in M9 minimal medium, pH 9.5 (NaOH adjusted), at 30 °C under aerobic conditions in an orbital shaker at 220 rpm [33]. Sodium acetate (50 mM) and ammonium chloride or sodium cyanide (2 mM) were used as carbon and nitrogen sources, respectively. When applicable, sodium arsenite or sodium arsenate dibasic (Sigma-Aldrich, St. Louis, MO, USA) was added at different concentrations from stock solutions. When agar plates were required, 15 g/L bacteriological agar was added to the liquid LB medium.

### 3.2. Determination of Bacterial Growth and Arsenic Tolerance

Cell growth was monitored in triplicate liquid cultures by measuring the absorbance at 600 nm (A_600nm_) in a spectrophotometer. On agar plates, growth was determined by counting colony-forming units (CFU) with the drop plate technique [65].

Tolerance of *P. pseudoalcaligenes* CECT 5344 to arsenite and arsenate was determined by calculating the minimal inhibitory concentration (MIC) and the minimal bactericidal concentration (MBC) [66], using cultures in M9 liquid medium inoculated in U-shaped 96-well microtiter plates (in quintuplicate). After incubation at 30 °C with shaking at 220 rpm for 48 h, MIC was estimated as the lowest As concentration that inhibits visible growth as a dot at the bottom of the well, and MBC was calculated as the lowest As concentration that kills the bacteria, checked by CFU counting. To determine MIC and MBC for As(III), ammonium or cyanide (2 mM) was used as a nitrogen source with increasing concentrations of arsenite up to 75 mM. For As(V), the concentration range was increased up to 2.5 M. The influence of As(III) in the cyanide tolerance was determined by calculating the MIC and MBC values for cyanide (with increasing concentrations of cyanide up to 50 mM) in the absence or the presence of 0.25 mM arsenite.

### 3.3. Analytical Determinations and Biofilm Quantification

Extracellular ammonium was determined using the Nessler reagent according to the previously described colorimetrical method [67]. Cyanide was measured using chloramine T, barbituric acid, and pyridine reagents, as previously described [68]. Protein was quantified by the dye-binding method [69].

Biofilm production in *P. pseudoalcaligenes* CECT 5344 was quantified with crystal violet (absorbance at 570 nm) into 96-well microtiter plates in triplicate, taking into consideration the ‘edge effect’ [70]. Biofilm formation was normalized by protein quantification to avoid possible differences in growth derived from the nitrogen source used. For this purpose, the same number and position of wells were used for a parallel protein determination [69]. Statistical significance was analyzed by applying a two-tailed *t*-test analysis (Benjamini-Hochberg corrected) on the normalized protein crystal violet measurements.

Determination of intracellular arsenic was carried out by harvesting 100 mL of cells cultured in M9 minimal medium with 0.25 mM arsenite, using either 2 mM ammonium chloride (condition named NAs) or 2 mM sodium cyanide (condition named CNAs) as the sole nitrogen source. After centrifugation (10,000 rpm, 4 °C, 10 min), cells were resuspended and washed three times with 1 mL 0.85% NaCl and dried at 80 °C for 96 h to determine cell dry weight. Then, pellets were resuspended in 1 mL HNO_3_ (69%, trace-metal grade, Fisher). On the other hand, the determination of arsenic present in the biofilm was carried out in cultures on microtiter plates, using six wells per replica, which were washed with 0.85% NaCl after growing for 48 h in the conditions NAs and CNAs. Then, 100 μL of HNO_3_ was used to collect biofilm. In parallel, the same number of wells for each condition were used for protein quantification [69]. Samples from both approaches (intracellular and biofilm) were heated up to 95 °C and digested at 25 °C for 1 h with shaking, and after dilution of the samples in 2% HNO_3_, arsenic content was analyzed on an ICP/MS equipment (PerkinElmer, model Nexion 350X) at the Central Service for Research Support of the University of Córdoba (SCAI-UCO). Three biological samples were analyzed for each condition (NAs and CNAs). Statistical significance was analyzed by applying a two-tailed *t*-test analysis (Benjamini-Hochberg corrected).

### 3.4. Quantitative Proteomic Analysis

Proteomic analysis was carried out in cells grown in M9 minimal medium under four conditions: 2 mM ammonium as sole nitrogen source, without arsenite (condition N) or with 0.25 mM arsenite (condition NAs); and 2 mM cyanide as sole nitrogen source, without (condition CN) or with 0.25 mM arsenite (condition CNAs). Before total consumption of the nitrogen source, three independent cultures were harvested by centrifugation, and pellets were resuspended in 300 μL of a lysis buffer that contained 50 mM Tris-HCl (pH 7.5), 4% CHAPS, and urea 8 M. Then, samples were disrupted by sonication in a Bandelin Sonoplus H2070 equipment (8 pulses for 20 s, at 25 W). Cell debris was removed by centrifugation (12,000 rpm, 10 min, 4 °C), and supernatants were precipitated using the 2D-Clean Up Kit (Amersham GE Healthcare). Pellets resuspended in 100 μL lysis buffer were used for protein determination [69] and LC-MS/MS analysis, as previously described [71,72]. The search against P. pseudoalcaligenes CECT 5344 proteome (UniProt UP000032841) and quantification parameters used for the proteomic analysis are shown in Appendix A. Data were analyzed using the Perseus software (1.6.12.1) (https://maxquant.org/perseus/ (accessed on 8 March 2023), and their statistical significance was determined by *t*-test analysis (Benjamini-Hochberg corrected). Proteins were considered differentially expressed when the fold change (FC), defined as the ratio of the normalized peptide intensity values in the non-reference condition vs. the reference condition, was ≥2 (for over-represented proteins) or ≤0.5 (for down-represented proteins), with an adjusted *p*-value < 0.05. The gene ontology GO enrichment analysis was carried out by using the Comparative GO application [73]. Data were deposited to the ProteomeXchange Consortium (http://proteomecentral.proteomexchange.org (accessed on 8 March 2023)) with the dataset identifier PXD033539.

### 3.5. Quantitative Real-Time PCR Analysis

The transcriptional expression of several selected genes was analyzed by qRT-PCR using three biological samples, each with two technical replicas, as previously described [72]. Gene-specific primers were designed using the Oligo 7.0 software (Appendix A). Data were normalized to the rpoB and 23S rRNA housekeeping genes. Finally, data were analyzed by a *t*-test (Benjamini-Hochberg corrected), and relative quantification was calculated by the ΔΔCt method [74].

## 4. Conclusions

The following conclusions may be addressed: (1) P. pseudoalcaligenes CECT 5344 contains two main *ars* gene clusters that respond to arsenite, regardless of the nitrogen source used, although As-resistance response shows some differences between ammonium and cyanide; (2) the main basis of As-resistance in this bacterium is the extrusion of As(III) through ArsBA efflux pumps, or as organic derivatives exported through ArsP, Acr3, and ArsJ permeases; (3) production of arseno-phosphoglycerate by the As-inducible GAPDH enzyme, As methylation, THF metabolism, and perhaps formation of Lewis adducts in the presence of cyanide, could be also relevant processes involved in As detoxification; (4) after extrusion of As(III) or organoarsenicals, As is accumulated in the bacterial biofilm, whose synthesis is enhanced in the presence of arsenite; (5) the ArsH2 protein may protect from the oxidative stress caused by both arsenic and cyanide; and (6) P. pseudoalcaligenes CECT 5344 could be used for bioremediation of wastes from mining and other industries co-contaminated with cyanide and arsenic.

## Figures and Tables

**Figure 1 ijms-24-07232-f001:**
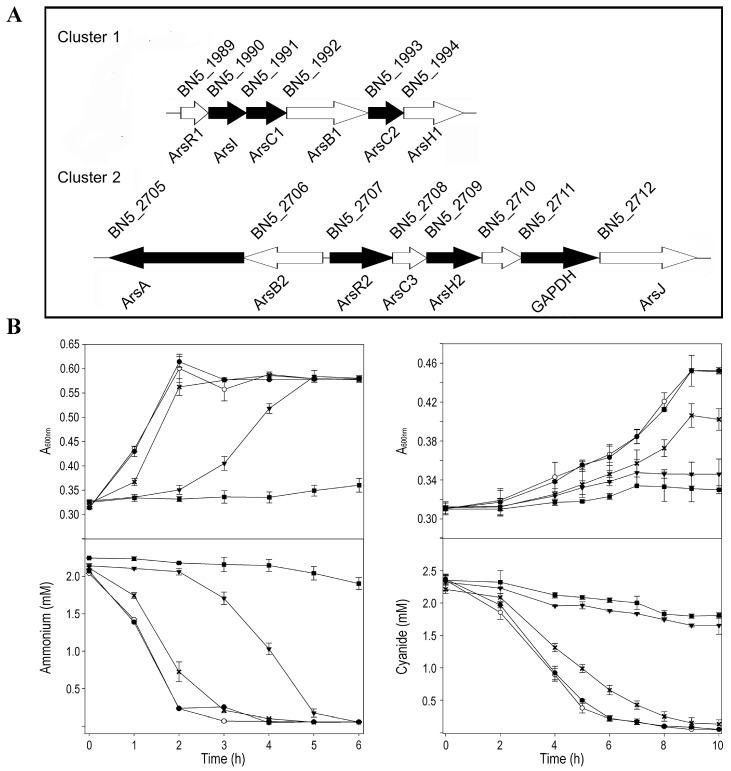
*P. pseudoalcaligenes* CECT 5344 *ars* gene clusters and physiological characterization of arsenic tolerance. (**A**) Genes coding for putative proteins related to arsenic detoxification. The reference numbers correspond with UniProt annotation (UP000032841). Black genes code for proteins identified by LC-MS/MS proteomic analysis in cells grown in the presence of arsenite. Gene descriptions are provided in Appendix A. (**B**) Arsenite tolerance of the strain CECT 5344. Cells were grown in M9 minimal medium with 50 mM sodium acetate as carbon source and 2 mM ammonium chloride (panels left) or 2 mM sodium cyanide (panels right) as nitrogen source, without (open circles) or with arsenite 0.1 mM (filled circles), 0.25 mM (asterisks), 0.5 mM (filled triangles), and 1 mM (filled squares). Bacterial growth, upper panels; N-source consumption, lower panels.

**Figure 2 ijms-24-07232-f002:**
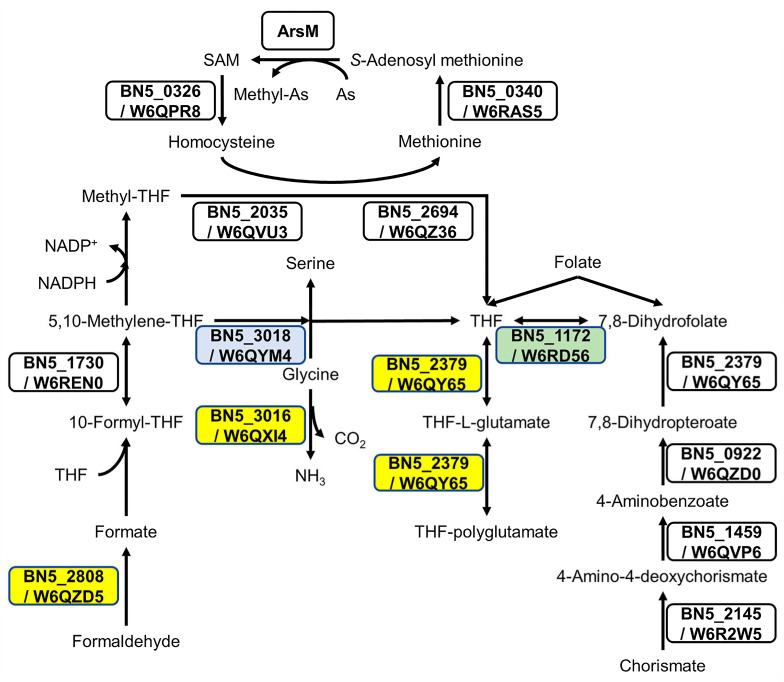
The effect of arsenite on proteins involved in tetrahydrofolate (THF) metabolism. Folate biosynthesis (ppse00790), biosynthesis of cofactors (ppse01240), and carbon metabolism (ppse01200) from KEGG were analyzed to build the metabolic pathways. Proteins W6QZD5 (*S*-formylglutathione hydrolase encoded by BN5_2808), W6QXI4 (glycine cleavage system H protein encoded by BN5_3016), and W6QY65 (dihydrofolate/folylpolyglutamate synthase encoded by BN5_2379), marked in yellow boxes, were induced by As(III) when ammonium was used as N-source. The protein W6QYM4 (serine methylase encoded by BN5_3018), shown in a blue box, was induced by As(III) with cyanide as N-source. The protein W6RD56 (dihydrofolate reductase encoded by BN5_1172), boxed in green, was identified only in cyanide, with or without arsenite.

**Figure 3 ijms-24-07232-f003:**
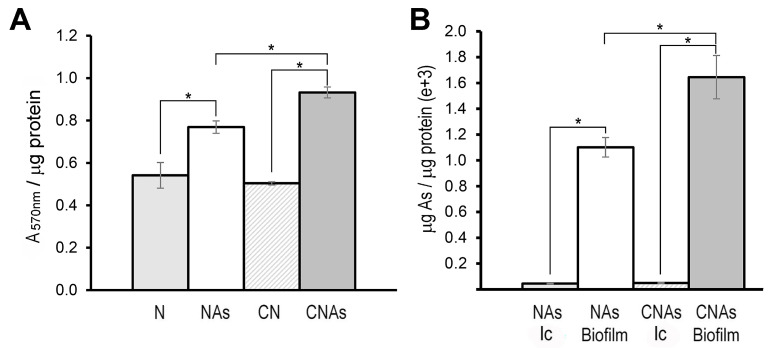
Biofilm production in *P. pseudoalcaligenes* CECT 5344 and arsenic accumulation in the biofilm. (**A**) Biofilm formation (measured with crystal violet as A_570nm_) normalized by protein amount (μg) in cells grown with 2 mM ammonium as a nitrogen source in the absence (N) or the presence of 0.25 mM arsenite (Nas), or with 2 mM cyanide as nitrogen source, without (CN) or with 0.25 mM arsenite (CNAs). (**B**) Intracellular (Ic) and biofilm As accumulation in cells grown with 0.25 mM arsenite using ammonium (Nas) or cyanide (CNAs) as the sole nitrogen source. Asterisks indicate that the differences are significant (*p <* 0.05) according to a *t*-test.

**Figure 4 ijms-24-07232-f004:**
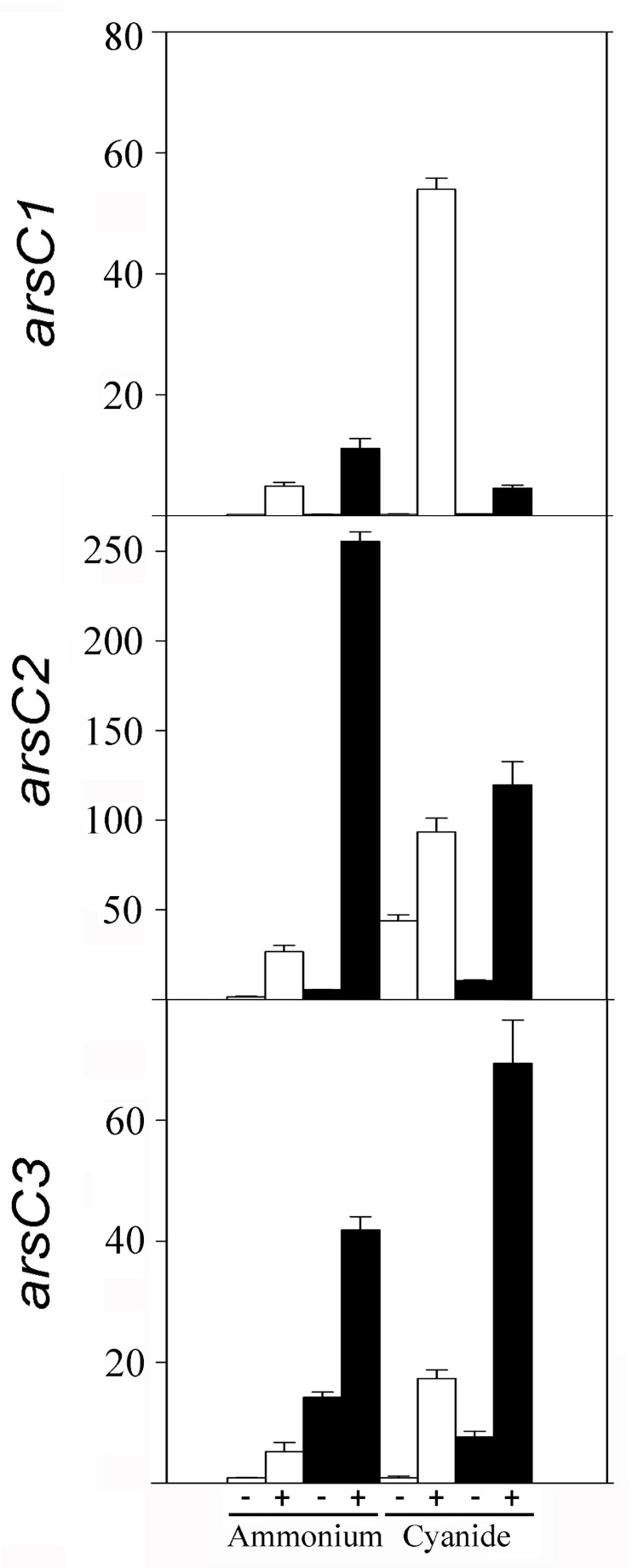
Relative expression of arsenate reductase genes (*arsC1*, *arsC2*, and *arsC3*) of P. pseudoalcaligenes CECT 5344 in response to arsenic. Gene expression was analyzed by qRT-PCR in cells grown with 2 mM ammonium chloride or 2 mM sodium cyanide as the sole nitrogen source, without (−) or with (+) arsenite 0.25 mM (white bars) or arsenate 200 mM (black bars).

**Figure 5 ijms-24-07232-f005:**
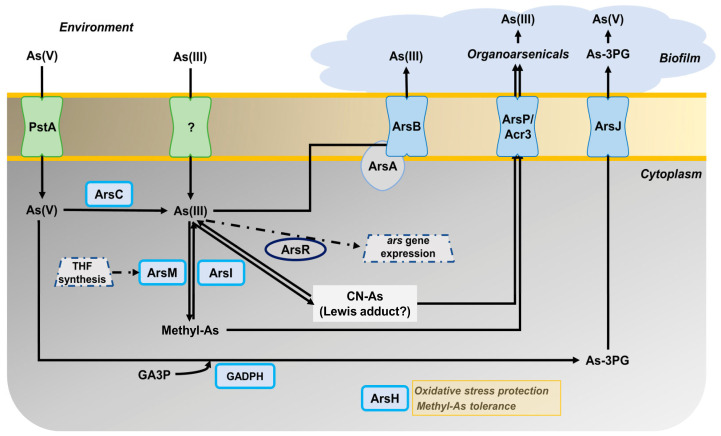
Overview of the arsenic metabolism in P. pseudoalcaligenes CECT 5344. The different proteins (encoded by the genes indicated in brackets) that are involved in the As tolerance of the strain CECT 5344 are: ArsC (BN5_1991; BN5_1993; BN5_2708); ArsR (BN5_1989; BN5_2707); ArsB (BN5_1992; BN5_2706); ArsA (BN5_2705); ArsI (BN5_1990); ArsH (BN5_1994; BN5_2709); glyceraldehyde-3-phosphate dehydrogenase GAPDH (BN5_2711); ArsJ (BN5_2712); ArsP (BN5_1279); Acr3 (BN5_2698); ArsM (BN5_1671; BN5_0407; BN5_4176); PstA (BN5_0117).

**Table 1 ijms-24-07232-t001:** Proteomic analysis of proteins with a possible role in arsenic and/or cyanide detoxification. Relative label-free quantification (LFQ) intensity was used to calculate the FC in the four comparisons NAs/N, CNAs/CN, CN/N, and CNAs/NAs. NAs, ammonium as N-source, with 0.25 mM arsenite; N, ammonium as N-source, without arsenite; CNAs, cyanide as N-source, with 0.25 mM arsenite; and CN, cyanide as N-source, without arsenite. Proteins exclusively found in a condition are indicated with the name of that condition. Proteins not detected (-).

Protein ID	Locus	Protein Names	Short Name	FC NAs/N	FC CNAs/CN	FC CN/N	FC CNAs/NAs
***ars* gene cluster 1**							
W6QX55	BN5_1989	Putative transcriptional regulator ArsR	ArsR1	-	-	-	-
W6RF96	BN5_1990	Lactoylglutathione lyase, C-As lyase	ArsI	NAs	CNAs	-	0.15
W6QUH3	BN5_1991	Arsenate reductase (thioredoxin-dependent)	ArsC1	15.27	3.21	3.01	0.63
W6R2E3	BN5_1992	Arsenical-resistance protein (arsenite ArsB efflux pump)	ArsB1	-	-	-	-
W6QVQ7	BN5_1993	Arsenate reductase (glutaredoxin-dependent)	ArsC2	3.21	CNAs	-	0.3
W6QX59	BN5_1994	Arsenical resistance protein	ArsH1	-	-	-	-
***ars* gene cluster 2**							
W6RHD6	BN5_2705	Arsenical pump-driving ATPase, ArsA	ArsA	NAs	2.38	CN	0.72
W6QWP7	BN5_2706	Arsenite transporter ArsB efflux pump	ArsB2	-	-	-	-
W6R4K3	BN5_2707	Regulatory protein ArsR	ArsR2	NAs	8.98	CN	0.67
W6QXT2	BN5_2708	Arsenate reductase (thioredoxin-dependent)	ArsC3	-	-	-	-
W6QZ50	BN5_2709	Arsenical resistance protein, ArsH	ArsH2	NAs	3.88	CN	0.99
W6RHD9	BN5_2710	Protein tyrosine phosphatase domain-containing protein 1		-	-	-	-
W6QWQ1	BN5_2711	Glyceraldehyde-3-phosphate dehydrogenase	GADPH	8.49	2.00	2.16	0.51
W6R4K6	BN5_2712	MFS transporter 1-arseno-3-phosphoglycerate exporter	ArsJ	-	-	-	-
***nit1C* gene cluster**							
H9N5E0	BN5_1630	Sigma-54-dependent transcriptional regulator	NitA	-	0.8	CN	CNAs
H9N5E2	BN5_1631	Uncharacterized protein	NitB	1.66	1.03	56.13	34.72
H9N5E1	BN5_1632	Nitrilase NitC	NitC	2.29	0.91	769.53	305.89
H9N5E3	BN5_1633	Radical SAM domain-containing protein	NitD	0.32	0.13	164.02	67.29
H9N5E4	BN5_1634	GCN5-related N-acetyltransferase	NitE	NAs	1.83	CN	30.76
H9N5E5	BN5_1635	AIR synthase-like protein	NitF	-	0.95	CN	CNAs
H9N5D9	BN5_1636	Uncharacterized protein	NitG	-	0.35	CN	CNAs
H9N5D8	BN5_1637	FAD dependent oxidoreductase	NitH	1.32	0.94	74.27	52.98
***cio* gene cluster**							
W6QWX1	BN5_1899	GntR family transcriptional regulator	MocR	-	CN	CN	-
W6RF17	BN5_1900	Sulfite reductase (NADPH) hemoprotein beta-component	CysL3	0.75	0.09	17.44	2.05
W6QU90	BN5_1901	Uncharacterized protein	CioC3	NAs	0.22	CN	30.15
W6R254	BN5_1902	Terminal oxidase subunit I	CioA3	-	CN	CN	-
W6QVH5	BN5_1903	Cytochrome *d* ubiquinol oxidase, subunit II	CioB3	-	-	-	-
W6QWX6	BN5_1904	Phosphoserine aminotransferase	SerC3	0.45	0.21	53.62	25.19
W6RF21	BN5_1905	Histidinol-phosphate aminotransferase	HisC3	0.59	0.22	33.33	12.33
W6QU95	BN5_1906	Acetylornithine aminotransferase	ArgD3	0.91	0.11	399.11	47.46
W6R260	BN5_1907	4-hydroxy-tetrahydrodipicolinate synthase	DapA1	2.11	0.3	182.98	26.06
W6QVI0	BN5_1908	High-affinity glucose transporter	-	-	-	-	-
W6QWY1	BN5_1909	Methylenetetrahydrofolate reductase	MetF3	-	0.13	CN	CNAs
W6RF25	BN5_1910	Cysteine synthase	CysM3	-	-	-	-
W6QUA1	BN5_1911	NADP-dependent malic enzyme	MaeB3	1.76	0.08	104.5	4.8
W6R265	BN5_1912	Nitrilase Nit4	Nit4	1.35	0.29	6.15	1.34
**Other nitrilases and C-N hydrolases**							
W6RF39	BN5_1925	Aliphatic nitrilase	Nit1	NAs	1.65	CN	1.43
W6R989	BN5_4427	Bifunctional nitrilase/nitrile hydratase NIT4B	Nit2/Nit4B	-	1.15	CN	CNAs
W6QQS6	BN5_0258	C-N hydrolase, nitrilase/cyanide hydratase	C-N11/AguB	-	-	-	-
W6QQZ6	BN5_0736	C-N hydrolase, nitrilase/cyanide hydratase	C-N10/Nit1B	NAs	0.86	CN	1.4
W6QUY5	BN5_1196	Nitrilase homolog 1	C-N12/Nit1A	-	-	-	-
W6QZ84	BN5_2750	Hydrolase, carbon-nitrogen family	C-N12	-	-	-	-
W6QY63	BN5_3204	Formamidase	C-N2/AmiF	1.95	1.28	5.32	3.49
W6QZD7	BN5_3251	C-N hydrolase, nitrilase/cyanide hydratase	CN-13/Nit3B	1.43	1.02	2.07	1.49
***cyn* gene cluster**							
W6QRB2	BN5_0438	Fis family transcriptional regulator	CynF	NAs	CN	CN	NAs
W6QST4	BN5_0439	ABC-type transporter periplasmic component protein	CynA	0.6	0.54	16.22	14.5
W6RB18	BN5_0440	ABC transporter inner membrane subunit protein	CynB	-	-	-	-
W6QQ36	BN5_0441	ABC transporter/ATPase component protein	CynD	1.06	1.53	61.12	88.16
W6QY14	BN5_0442	Cyanase	CynS	0.45	0.87	164.76	318.2
**Other proteins**							
W6QRW0	BN5_0117	Phosphate transport system permease protein	PstA	NAs	1.41	CN	2.04
W6QXN5	BN5_0312	Methyl-accepting chemotaxis sensory transducer	PilJ	NAs	2.08	CN	9.80
W6QSH6	BN5_0328	Glutamate synthase	PydA	NAs	1.26	CN	0.82
W6QXQ0	BN5_0332	*N*-carbamoyl-L-amino acid amidohydrolase	-	NAs	1.37	CN	2.10
W6QTB1	BN5_1579	Molybdopterin molybdochelatase	MoeA	NAs	0.84	CN	2.73
W6QWV1	BN5_1877	Siroheme synthase	CysG	NAs	1.31	CN	3.94
W6QRZ0	BN5_1113	Glycerol-3-phosphate dehydrogenase	GlpD1	NAs	CNAs	-	1.89
W6QSH3	BN5_1279	UPF0718 protein MJ0584	ArsP	NAs	CNAs	-	1.51
W6QXS5	BN5_2698	Multidrug resistance protein MdtC, Acr3-type permease	Acr3/MdtC	NAs	CNAs	-	0.22
W6R038	BN5_3066	Two component LuxR family transcriptional regulator	LuxR	NAs	CNAs	-	2.78
W6R1S4	BN5_3676	Protein translocase subunit SecY	SecY	NAs	CNAs	-	4.28
W6R067	BN5_3092	Diguanylate cyclase YddV	YddV	-	CNAs	-	CNAs
W6RL26	BN5_4037	Putative membrane transport ATPase	ZntA	-	CNAs	-	CNAs
W6R8E4	BN5_4137	dTDP-4-dehydrorhamnose 3,5-epimerase	RfbC	-	CNAs	-	CNAs
W6R0W4	BN5_4181	Uncharacterized protein	-	-	CNAs	-	CNAs
W6RM49	BN5_4450	AraC family transcriptional regulator	AraC	-	CNAs	-	CNAs
W6QSN9	BN5_0908	DNA repair protein RecN	RecN	-	CNAs	-	CNAs
W6RCR6	BN5_1077	Putative acyl-CoA thioester hydrolase	-	-	CNAs	-	CNAs

## Data Availability

Appendix A associated with this work are available online, and it can be only found in the online version of this paper.

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
