# Peer review of "Proteomic Analysis of Arsenic Resistance during Cyanide Assimilation by Pseudomonas pseudoalcaligenes CECT 5344"

_ijms, 2023, doi:10.3390/ijms24087232_

Round 1

Reviewer 1 Report

This study focused on the arsenic metabolism of P. pseudoalcaligenes CECT 5344 under the cyanide assimilating condition. From the results of the proteomics and qRT-PCR analysis, the authors predicted the whole shape of the arsenic resistance mechanism. The results were very interesting. The scientific and writing qualities were sufficient. This work should be published on IJMS after a minor revision.

 According to the introduction, part of the objective of this study is to reveal how the coexistence of cyanide affects the arsenic metabolism. However, it is rather unclear in this manuscript. I’d like to recommend adding some discussions on the effect of cyanide on the arsenic metabolism from scientific and practical viewpoints. 

Author Response

  1. pseudoalcaligenes CECT 5344 may detoxify simultaneously cyanide and arsenic, and these bioremediation processes affect each other. To emphasize the relationship between cyanide and arsenic metabolism, some parts of the text have been rewritten and a new paragraph has been included. Arsenite causes a decrease in the expression of the cio gene cluster involved in cyanide resistance, as well as in the levels of the NitD and NitG proteins encoded by the nit1C gene cluster required for cyanide assimilation. This is currently indicated in lines 219-225 and 405-408. The negative effect of arsenic on the cyanide degradation process was also verified determining the MIC and MBC values for cyanide, in the absence or presence of arsenite. This is presented and discussed in lines 146-150. Likewise, the presence of cyanide could facilitate the formation of Lewis adducts with arsenic (lines 306-307 and 422-423). On the other hand, cyanide also affects arsenic metabolism by up-regulating ArsH2 and other Ars proteins, even in the absence of arsenite, as revealed by the comparison CN vs N in Table 1. This is discussed in lines 310-316, but a new paragraph highlighting the relationship between cyanide and arsenic detoxification processes has been included (lines 231-241). Also, a new sentence emphasizing this fact was included in lines 427-428.

Reviewer 2 Report

Authors developped some strategies for bioremediating wastewater contaminated with arsenic, which is interesting topic in environmental engineering.  I recommend it publication in IJMS. However, the manuscript should be perfected further before accepted for publication.

1) Please introduce where was Pseudomonas pseudoalcaligenes CECT 5344 taken from, and what characteristics in biodegradation of arisenic.

2) Please exlain clearly what relationship between Figure 1(a) and Figure 1(b), and which gene played key role in the arisenic.

3) What role did the phosphate play in arisenic biodegradation?

Author Response

1) Please introduce where was Pseudomonas pseudoalcaligenes CECT 5344 taken from, and what characteristics in biodegradation of arsenic.

The sentence "isolated from sludge of the Guadalquivir River (Córdoba, Spain)" has been added in lines 93-94. The role of this bacterium in the biodegradation of arsenic is addressed in this work, and it was not tested before.

2) Please explain clearly what relationship between Figure 1(a) and Figure 1(b), and which gene played key role in the arsenic.

Two different analyzes are shown in figures 1a and 1b. In Fig. 1a, the main genes involved in arsenic detoxification, which are arranged in two gene clusters, are indicated. In addition, those genes encoding proteins that were identified by proteomics are shown in black. On the other hand, Fig. 1b shows the growth and N-source consumption curves of the strain P. pseudoalcaligenes CECT 5344 with ammonium or cyanide, at different arsenite concentrations. The aim of this physiological characterization was to find the optimal culture conditions for the functional analyses (proteomics and qRT-PCR) to study the influence of arsenite. These figures could be shown as separate figures, but they were clustered for conciseness, shorting the paper, and because the capability to grow in the presence of arsenite (Fig. 1b) is eventually due to the proteins encoded by the ars genes shown in Fig. 1a.  

3) What role did the phosphate play in arsenic biodegradation?

It is well known that arsenate is a structural analogue of phosphate, and it enters the cell by using the Pst phosphate transporter. Therefore, although it is really an interesting question that could be addressed in further studies, the influence of different phosphate concentrations on arsenic biodegradation was not addressed in the present work. It would be expected, however, that at a high phosphate concentration, arsenate bioremediation would be diminished since both compounds would compete for the same pathway to enter the cell, as it has been demonstrated in plants (Zvobgo et al. The effects of phosphate on arsenic uptake and toxicity alleviation in tobacco genotypes with differing arsenic tolerances. Environ Toxicol Chem. 2015 Jan;34(1):45-52. doi: 10.1002/etc.2776; Sneller et al. Toxicity of arsenate in Silene vulgaris, accumulation and degradation of arsenate-induced phytochelatins. New Phytol. 1999, 144:223–232).